# Chickpea-Based Milk Analogue Stabilized by Transglutaminase

**DOI:** 10.3390/foods14030514

**Published:** 2025-02-05

**Authors:** Barak Snir, Ayelet Fishman, Jovana Glusac

**Affiliations:** 1Department of Biotechnology and Food Engineering, Technion-Israel Institute of Technology, Haifa 3200003, Israel; bsnir@campus.haifa.ac.il (B.S.); afishman@bfe.technion.ac.il (A.F.); 2BioSense Institute—Research and Development Institute for Information Technologies in Biosystems, University of Novi Sad, Dr. Zorana Đinđića 1a, 21000 Novi Sad, Serbia

**Keywords:** chickpea protein, plant-based milk analogues, transglutaminase, stability, shelf-life

## Abstract

Plant-based milk substitutes are becoming increasingly popular in the food industry. Among different plant proteins, chickpea proteins (CP) offer unique qualities as good functional and nutritional properties, followed by pleasant taste. This study examines the ability of the production of *o*/*w* emulsions resembling milk analogue (3% *w*/*w* chickpea protein, 3% *w*/*w* canola oil) by using chickpea protein isolate with/without the enzyme transglutaminase (TG) (50 U/g of protein). As a reference material, commercial soymilk was used. The emulsions were characterized by particle size distribution, zeta potential, viscosity, and microstructure. The TG-crosslinked chickpea protein milk analogue demonstrated improved stability, characterized by enhanced zeta potential (−24.7 mV) and extended shelf life compared to chickpea protein milk analogue without TG and soymilk. Stable particle size distribution (D[3,2] 0.11–0.17 µm) and shear-thinning behaviour (viscosity values of 2.16 mPas at 300 1/s) additionally contributed to their stability and desirable viscosity. Overall, chickpea protein milk analogue crosslinked by TG presents a promising alternative to traditional and plant-based milk products, offering clean-label, functional, and shelf-stable formulations. The additional optimization of protein concentration and processing conditions could enhance the overall functionality even further.

## 1. Introduction

The growing population and climate change are putting increasing pressure on natural resources. Meeting global food demands requires not only more food but also diverse, nutritious options. A significant challenge is the rising need for functional proteins, as animal-based foods—currently the main protein source in the Western diet—contribute essential nutrients but have a substantial environmental footprint. Plant sources of protein offer a sustainable alternative, providing health benefits. Chickpea stands out as the third most cultivated legume worldwide, boasting elevated protein content (14.9–24.6%), essential amino acids, and low water and fertilizer requirements, making it a promising solution for addressing global food security [1,2].

Over recent years, there has been a surge in consumer interest in plant-based food products, driven by the increasing demand for healthier and more sustainable lifestyles. This trend is particularly evident in the beverage sector, where plant-based milk alternatives are gaining popularity. These alternatives provide solutions for individuals with dietary restrictions, such as cow milk allergy or lactose intolerance, and also offer health-conscious alternatives to traditional dairy milk, addressing concerns about calories, cholesterol, and fat intake [3,4,5]. With the global demand for milk and meat projected to increase by 57% and 48%, respectively, until 2050, followed by intensifying water, land, and greenhouse gas challenges, the need for alternative protein sources will become even more critical [4,6]. Furthermore, regions such as South America, Asia, and Africa are home to populations where lactose intolerance affects over 50% of people, further driving the demand for lactose- and cholesterol-free plant-based milk options [7,8]. The expanding market, which surpassed $15 billion in 2023, is projected to grow at a CAGR of 11.6% from 2024 to 2032 https://www.gminsights.com/industry-analysis/plant-milk-market (accessed on 1 November 2024). This expanding market will continue to benefit from the diversity of plant-based proteins.

Proteins play a crucial role in the food industry as emulsion stabilizers, thanks to their ability to adsorb at oil–water interfaces and form protective layers around oil droplets [9]. During emulsion formation, proteins create a densely packed layer on newly formed droplets, reducing interfacial tension and preventing coalescence by acting as a physical barrier [10]. Stability is further enhanced by electrostatic repulsion when the pH is far from the protein’s isoelectric point (pI) and by steric hindrance, as protein segments extend into the aqueous phase, restricting droplet interaction. Proteins also increase the viscosity of the continuous phase, slowing droplet movement and enhancing stability [11]. Plant-based proteins, in particular, have gained attention for their ability to stabilize oil–water interfaces, leveraging their amphiphilic properties, offering a natural and clean-label alternative to synthetic emulsifiers [4,5,11]. This makes them ideal for use in food products targeting health-conscious and environmentally aware consumers who are increasingly looking for sustainable and allergen-friendly options [4].

Crosslinked proteins offer even greater stabilization and texture enhancement in emulsion [12]. Crosslinking increases electrostatic repulsion and forms aggregates around droplets, while also imparting gel-like properties to the continuous phase [13,14]. Crosslinking can be achieved via physical, chemical, or enzymatic methods. Physical methods, like heating or high-pressure treatment, are safe and often combined with other techniques [15]. Chemical crosslinking formed with simple reagents raises toxicity concerns [16]. Enzymatic methods, by contrast, are preferred for their specificity, efficiency, and safety [12]. Advanced commercial enzymes resist pH, salt, and temperature fluctuations, making them highly effective [17]. Enzyme transglutaminase (TG) catalyzes acyl-transfer reactions between a glutamine residue (acyl donor) and a primary amine (such as lysine) in proteins, forming ε-(γ-glutamyl) lysine isopeptide bonds [18]. This reaction generates ammonia and, in the absence of suitable amine substrates, can lead to deamination by using water as the acyl acceptor [18]. While TG has specific glutamine substrate requirements, it demonstrates broad specificity for acyl acceptor substrates, making it versatile for food applications [19].

The objective of this study was to explore the potential of microbial TG for crosslinking chickpea protein in the production of chickpea-based milk analogues. The rational was to enhance the emulsifying properties of the protein, improving the stability of oil–water emulsions and extending the shelf life of the produced drink. Canola oil was selected as a cost-effective and high-quality oil source for producing the plant-based milk analogue. Its advantages are neutral taste and smell, as well as a moderate smoking point. This study also examined a comparison in the performance of chickpea-based milk analogues with TG to those made without TG and to a commercial reference soy-derived drink. The obtained emulsions were characterized by particle size distribution, zeta potential, viscosity, and microstructure. The results of this research could contribute to the development of more stable, nutritious, and sustainable plant-based milk alternative products, addressing the growing consumer demand for functional and allergen-free food options.

## 2. Materials and Methods

Materials

Canola oil, chickpea seeds, and soymilk (Soy drink lite, Tnuva alternative, 3.3% protein, 1.6% fat, sugar, dietary supplement: calcium carbonate (of which calcium 0.1%), acidity regulators: E-450(v), E-452(i), E-500(i), flavourings, salt, stabilizer: gellan gum, vitamin B12, B2 and D (B2—0.24 mg, B12—0.4 mcg, D—0.75 mcg per 100 g) were sourced from a local supermarket in Haifa, Israel. ACTIVA^®^TI TG, derived from the bacterial source *Streptoverticillium mobaraense* (strain S-8112) was obtained from Ajinomoto (Ajinomoto Food Ingredients LLC., Chicago, IL, USA). Chemicals including Bradford reagent, NaOH, HCl, Tris, β-mercaptoethanol, Coomassie brilliant blue (R-250), ethanol, acetic acid, and methanol were purchased from Sigma Chemical Co. (Rehovot, Israel).

Isolation of chickpea protein fractions

Fresh chickpea seeds were ground, and chickpea protein was isolated using isoelectric precipitation as described in a previous study [11]. A total of 100 g of milled chickpea flour was suspended in distilled water (DW) at a 1:10 ratio (*w*/*v*). The pH of mixture was adjusted to 9 using 2M NaOH and it was stirred at 500 rpm for 90 min at room temperature. The resulting suspension was centrifuged at 4500× *g* for 20 min at 4 °C using a Thermo Scientific™ Sorvall™ LYNX 4000 (Thermo Fisher Scientific, Waltham, MA, USA), and the supernatant was subsequently collected. The pellet was resuspended in DW at a ratio of 1:5 (*w*/*v*). Then, again the suspension was adjusted to pH 9 and centrifuged under the same conditions at 4500× *g* for 20 min at 4 °C. Both supernatants were pooled and adjusted to the isoelectric point (pH 4.6) with 1 M HCl, in order to precipitate the protein fraction. Afterwards, the mixture was centrifuged at 8000× *g* for 20 min at 4 °C. The supernatant was discharged, and the precipitate was dissolved in distilled water and adjusted to pH 7. The samples were dialyzed against water and freeze-dried. Total protein was evaluated by the Bradford method [20] and was in the range of 84–88%.

Emulsion preparation resembling milk

The oil-in-water emulsions resembling milk (3% fat, 3% protein) were produced using chickpea protein and canola oil, with or without the addition of enzyme TG. Four percent (*w*/*w*) of chickpea protein was dissolved overnight in distilled water at 4 °C under constant stirring. The following day, the protein dispersion was centrifuged at 1000× *g*/5 min/room temperature to remove large protein aggregates. The supernatant was adjusted to a protein concentration of 3% (*w*/*w*) and used for subsequent experiments. The soluble chickpea protein solution was heat treated at 85 °C/20 min to simulate the high-pasteurization process and ensure the death of most vegetative microorganisms [21]. Based on the previous study [14], with some modification, the enzyme concentration and incubation conditions were chosen. After cooling, the enzyme TG was added (50 U/g TG per protein weight) and incubated at 37 °C/3 h. A control sample without TG addition was produced under identical conditions. Canola oil in 3% (*w*/*w*) was added to the protein solution, and the mixture was homogenized using a shear dispersing unit (Pro200, Pro-Scientific Inc., Oxford, CT, USA) for 1 min at 35,000 rpm to produce a coarse emulsion. Fine emulsions were produced using high-pressure homogenization (EmulsiFlex-C3, Avestin Inc., Ottawa, ON, Canada) with 4 passes at 20 kPsi. The enzyme inactivation for both emulsions was carried out at 90 °C for 5 min. The commercial soymilk, serving as a reference, and the chickpea-based milk analogues were stored at room temperature (RT) for over a month to assess visual stability. Additionally, the plant-based milk analogues were stored at 4 °C for a month to facilitate all analytical evaluations.

ζ-Potential

The ζ-potential of the samples was measured using Zetasizer Ultra (Malvern Instruments, Worcestershire, WR14 1XZ, UK) following the method described previously [22]. Chickpea-based milk analogues and soymilk samples were diluted 1000-fold in distilled water and the particle surface charge potential was calculated using the Smoluchowski model.

Determination of particle size distribution (PSD)

Particle size was analyzed using MasterSizer 3000 laser diffraction particle size analyzer (Malvern Instruments Ltd., Malvern, Worcestershire, WR14 1XZ, UK) equipped with a wet sample dispersion unit (Malvern Hydro MV, Worcestershire, WR14 1XZ, UK). The background and sample integration times were set to 20 and 10 s, respectively. The optical properties were defined with a refractive index of 1.46 for canola oil and 1.330 for the dispersant (water), along with an absorption index of 0.001.

Viscosity

Dynamic viscosity of the emulsions was measured using a Discovery Hybrid Rheometer (DHR-2, TA Instruments, New Castle, DE, USA) equipped with parallel plates (d = 60 mm). Chickpea-based milk analogous, and soymilk were placed between parallel plates at a controlled temperature of 25 °C, with a plate gap of 1.0 mm. The sample temperature was regulated via the lower plate, and excess material was removed prior to measurements. The rheometer was operated using the Trios Express software (TA Instruments, New Castle, DE, USA) https://www.tainstruments.com/trios-software/ (accessed on 23 January 2025). The shear rate was increased from 0.5 to 300 1/s, and apparent viscosity was recorded as a function of shear stress after 1st and 28th day of storage.

Direct observation of emulsions

Images were acquired and processed using the ZEN lite image analysis software (Zeiss) https://www.zeiss.com/microscopy/en/products/software/zeiss-zen-lite.html (accessed on 23 January 2025). The microstructural analyses of chickpea-based milk analogues and soymilk were performed using a light microscope (BX51P, Olympus, Tokyo, Japan) in the bright-field mode. Images were taken at RT using an Olympus DP71 digital camera. Representative images were shown (n = 3 independent experiments).

Experimental design and analysis

All experiments were performed in triplicate, and the data are presented as the mean ± standard deviation. Statistical analysis was conducted using the SigmaPlot software package (Version 11.0, Systat Software Inc., San Jose, CA, USA) with a primary focus on paired sample t-tests under the assumption of equal variances. A significance threshold of *p* < 0.05 was applied.

## 3. Results

Pictures of plant-based milk analogues, chickpea-based milk analogues with and without TG addition and commercial soymilk were taken to assess visual stability during 28 days of storage at 4 °C and room temperature (RT) (Figure 1). After the 7th day of storage at RT, a layer of creaming was observed in chickpea-based milk analogues without TG addition and soymilk samples. In addition, the soymilk samples showed contamination after the 7th day of storage at RT, unlike milk analogues produced from chickpea protein. No difference was observed between samples stored at 4 °C, indicating a good stability under refrigerated conditions.

Particle size distribution (Figure 2, Table 1), zeta potential (Figure 3), dynamic viscosity (Figure 4), and emulsion microstructure (Figure 5) were evaluated in plant-based milk analogues during the storage time at 4 °C.

According to Figure 2, there were no significant differences in particle size distribution between chickpea-based milk analogues and the storage time, respectively. The PSD in chickpea-based milk analogues showed monomodal distribution over the storage period. The commercial soymilk exhibited a uniform monomodal particle size distribution over a 7-day storage period. After the 7th day of storage, the Ostwald ripening phenomenon was observed in commercial soymilk, indicating the coalescence of particles. On the other hand, the chickpea-based milk analogues regardless of TG addition showed stable PSD without any changes observed during the storage time. The stability of an emulsion is intricately tied to the particle size and ζ-potential of the solution. In addition, the key parameters characterizing particle size distribution (PSD) are the volume-weighted mean diameter D[4,3] and the surface area-weighted mean diameter (D[3,2], also known as the Sauter mean diameter. D[4,3] is particularly sensitive to the presence of larger particles, whereas D[3,2] is more influenced by smaller particles [23]. In plant-based milk analogues, the surface area-weighted mean diameter D[3,2] remained between 0.09 and 0.19 µm over 28 days of storage (Table 1). A more pronounced change was observed in the volume-weighted mean diameter D[4,3], with an increase in PSD occurring after day 21 in chickpea-based milk analogues, while in soymilk, this increase was detected as early as day 14 (Table 1) due to bimodal distribution and the appearance of larger particles.

Zeta potential was measured periodically over the 28 days of storage at 4 °C in all samples as shown in Figure 3. A negative charge was observed, particularly in the chickpea-based milk analogue with TG addition, which exhibited more than 2-fold higher ζ-potential values compared to its counterpart without TG and commercial soymilk. This result indicates enhanced and prolonged stability in the TG-treated sample. At the end of the storage period, the measured ζ-potential in the chickpea-based milk analogue with TG addition was the highest at −24.7 mV, followed by the sample without TG addition, −12.4 mV. The lowest value was measured in commercial soymilk with −7.8 mV.

The rheological behaviour of plant-based milk samples was measured after the first day of production and after 28 days of storage at 4 °C (Figure 4). According to Figure 4, it can be observed that in all samples the viscosity decreased with the increase in shear rate. The chickpea-based milk analogues with and without TG demonstrated comparable rheological behaviour across the tested shear rate range, with all emulsions exhibiting distinct shear-thinning behaviour. At a low shear rate, the initial viscosity of the chickpea-based milk analogues with TG (65.28 mPas) was similar to commercial soymilk (62.22 mPas), and higher than the sample without TG (49.74 mPas) after the 1st day of storage. Over the storage period, commercial soymilk consistently exhibited higher viscosity than both chickpea-based emulsions. At a shear rate of 300 1/s, the viscosity of soymilk (5.73 mPas) surpassed that of the chickpea-based milk analogues (2.16 and 2.48 mPas, with and without TG addition) at the end of the storage period. Overall, a slight thickening effect was observed in all emulsions during storage.

The structure of plant-based milk analogues was observed by light microscope and the results are presented in Figure 5. The microstructure of commercial soymilk was different compared to chickpea-based milk analogues. There is noticeably less coalescence in chickpea-based milk analogues with TG compared to its counterpart without TG addition. This highlights the unique benefits of crosslinking in enhancing the product’s stability. After one month of storage at 4 °C, more pronounced aggregation became visible in the chickpea-based milk analogues with TG addition, probably due to the crosslinking of proteins. The microstructure of commercial soymilk was characterized by smaller particles, which correlate to results obtained for PSD (Figure 2), while after a month of storage at 4 °C cluster formation was observed, presenting the crystallization of fat particles. 

## 4. Discussion

Chickpea protein consists of globulin (legumin, vicilin, convicilin) albumin (enzymatic proteins, protease and amylase inhibitors, and lectins), prolamin, and glutelin fractions, described in our previous work [14]. Among chickpea protein fractions, vicilin with a molecular weight of 50 kDa, chickpea α- and β-legumin, lipoxygenase (∼92 kDa), and convicilin (70 kDa) were crosslinked rapidly within 2 h by TG as showed previously [14].

The characteristic creamy appearance of milk-type products is primarily attributed to the scattering of light by fat droplets and other colloidal particles [4,5]. In this study, chickpea-based milk analogues with TG addition exhibited prolonged stability at room temperature compared to the rest of samples (Figure 1). The upper layer of chickpea-based milk analogue without TG addition and commercial soymilk displayed evident stratification, indicating aggregation within the sample.

Heat treatment (HT) is a common pretreatment method for plant-based milks to enhance safety, palatability, and stability [13,24,25]. However, HT can induce protein aggregation and sedimentation, particularly in emulsions stabilized by globular proteins like chickpea vicilin [14,26]. Under optimized conditions, HT can promote crosslinked protein formation, enhancing emulsion stability [25]. The denaturation temperature of chickpea protein was determined to be above 90 °C, which aligns with findings reported in the literature [27,28], Therefore, it can be assumed that the high-pasteurization temperature or enzyme inactivation heat treatment used in this study did not cause protein denaturation. The chickpea-based milk analogues without TG addition exhibited some visually observed aggregation (Figure 1) and lower zeta potential (Figure 3). Interestingly, PSD did not differ significantly compared to the sample that was enzymatically crosslinked (Figure 2).

A challenge in plant-based milk substitutes is their typically low-protein content, a consequence of solid–liquid separation processes and the insolubility of plant proteins [4,5,29]. This issue is particularly notable in oat proteins, which are known for their low solubility [29]. In this study, the zeta potential of droplets in chickpea milk analogue with TG showed a higher negative charge (Figure 3). The charge on oil droplets coated with proteins significantly influences emulsion stability [30]. For instance, cow’s milk stabilized by casein proteins typically has a zeta potential ranging around or below −20 mV due to mineral composition in the aqueous phase of milk, which can affect the ionic strength of milk [31]. Notably, chickpea-based milk analogue with TG had a more negative zeta potential than other plant-based milks such as almond (−17 mV), hazelnut (−23.8 mV) [32], and coconut (−16 mV) [33]. Generally, dispersions with zeta potentials exceeding ±30 mV are considered stable [26,30]. Compared to the reference commercial soymilk, both chickpea-based milk analogues displayed higher zeta potential, likely contributing to their extended shelf life.

Emulsion stability is also influenced by particle size distribution (PSD), with smaller particle sizes generally corresponding to greater stability and better mouthfeel [4]. The surface area-weighted mean diameter D[3,2] of the protein-based milk analogues remained stable throughout storage. In contrast, the volume-weighted mean diameter D[4,3] began to increase on the 21st day of storage for chickpea-based milk analogues, while for soymilk, this increase was observed on the 14th day (Figure 2, Table 1). The differences in PSD may be attributed to variations in isolation and processing methods, as well as the presence of thickeners in soymilk. In homogenized cow’s milk, a typical monomodal PSD of around 1 μm or slightly below has been reported [34], similar to chickpea-based milk analogues and commercial soymilk. Lentil protein-based milk analogues are also reported to have larger particle sizes [35].

Viscosity is a critical quality parameter for dairy beverages; overly high viscosity impairs flowability, while low viscosity may promote sedimentation and flocculation of colloidal particles [4,5,36]. The apparent viscosity of commercial soymilks ranges from 1.2 to 9.9 mPa·s, influenced by the use of thickening agents [37]. Soy and almond milks boast extended shelf lives of up to 170 days at 4 °C, whereas oat (28 days), rice (12 days), and peanut milk (30 days) are more perishable [38]. In contrast, chickpea-based milk analogue in this study demonstrated a robust shelf life, exceeding 30 days at room temperature and at 4 °C. These emulsions also exhibited lower viscosity compared to cow’s milk [39]. The initial high viscosity of the crosslinked emulsion decreased progressively under shear, exhibiting shear-thinning behaviour likely due to the delicate structure of CP milk [40]. At higher shear rates (300 1/s), the viscosity of the chickpea-based milk analogue with TG addition decreased more significantly compared to its non-crosslinked counterpart.

Light microscopy revealed slight structural differences between soymilk and CP-based milk analogues (Figure 5). Soymilk showed larger particle size, which is in correlation with results obtained for particle size analysis by laser diffraction (Figure 2, Table 1). At the end of the storage time, thickening in soymilk can be attributed to interactions among ingredients, as commercial soymilk usually contains thickening agents such as carrageenan, xanthan gum, gellan gum (present in the product used here), and fibres [4,37]. In particular, carrageenan has been recognized as a concern for its safety because of an article published in 2001 by Joanne Tobacman, claiming that carrageenan may cause lesions or cancer in the gastrointestinal tract [41], but also based on recent research confirming the disruption of gut epithelial function [42]. Considering the microscopic images, the oil droplets in chickpea protein milk analogue without TG can be seen to exist in different sizes in the continuous phase due to coalescence phenomena. Furthermore, the microstructural characteristics described for the plant-based milk alternatives emulsions align with the other studies involving different plant proteins [25,35,36].

Overall, the mechanism of TG crosslinking in enhancing stability in chickpea-based milk analogues is multifaceted. Crosslinking modifies the protein network, contributing to reduced coalescence through the formation of an enhanced interfacial protein layer, increased electrostatic repulsion (ζ-potential of −24.7 mV) and improved protein–protein interactions [5,14,26]. The crosslinked proteins form a denser and more cohesive layer at the oil–water interface, providing a stronger physical barrier against coalescence [5,14]. Furthermore, crosslinking stabilizes protein–protein interactions within the continuous phase, which minimizes collision frequency, further preventing coalescence and phase separation [26].

## 5. Conclusions

This study highlights the potential of chickpea protein, a highly resource-efficient legume with exceptional protein content, to contribute to sustainable food systems while catering to growing consumer demand for health-conscious and allergen-free milk alternatives.

The chickpea-based milk analogues formulated with TG demonstrated superior stability and shelf life compared to its counterpart without TG, and to reference material, commercial soymilk. Enhanced zeta potential (−24.7 mV) contributed to improved emulsion stability by increasing electrostatic repulsion, which minimized flocculation and sedimentation. Consistent particle size distribution with D[3,2] between 0.11 and 0.17 µm and shear-thinning behaviour further supported the stability and desirable viscosity. Crosslinking played a crucial role in structurally reinforcing the emulsion, which reduced coalescence over time.

Overall, the enzymatic crosslinking of chickpea protein demonstrates great promise for producing chickpea-based milk analogues, offering clean-label, functional, and shelf-stable formulations. However, there are certain limitations to consider, including the production costs associated with enzymatic crosslinking and high-pressure homogenization, which could pose challenges for large-scale manufacturing. Additionally, comprehensive sensory evaluations and consumer studies are essential to evaluate the market acceptance of these formulations. Addressing these challenges and optimizing aspects such as protein concentration, TG usage, and processing conditions will further enhance performance. By meeting these objectives, chickpea-based milk analogues have the potential to emerge as a compelling, sustainable alternative within the rapidly growing plant-based food market.

## Figures and Tables

**Figure 1 foods-14-00514-f001:**
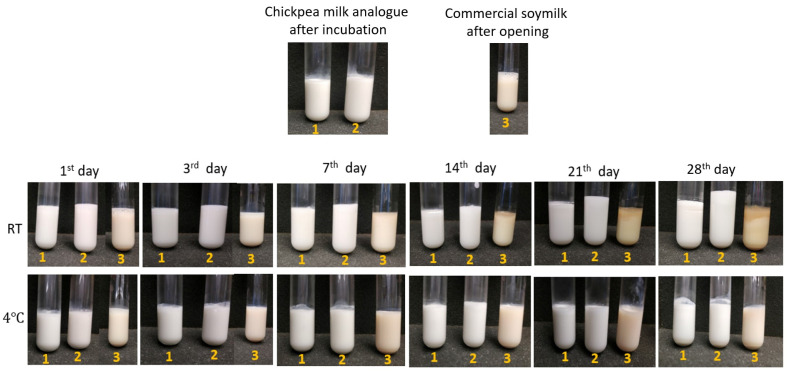
The visual appearance of chickpea-based milk analogues with TG (1), without TG addition (2), and commercial soymilk (3) over 28 days of storage time at 4 °C and room temperature (RT).

**Figure 2 foods-14-00514-f002:**
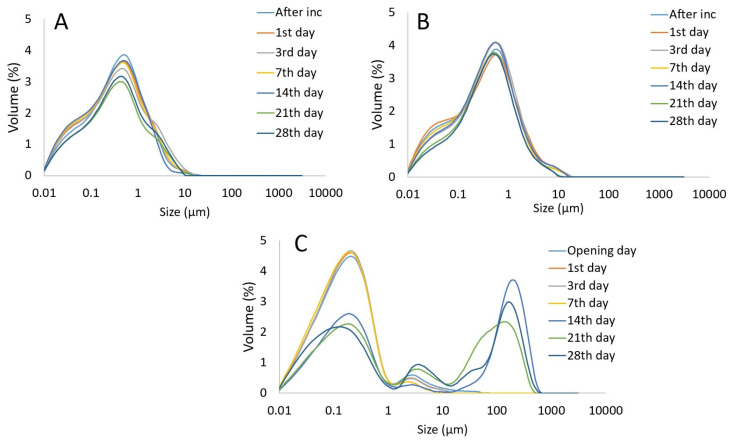
Particle size distribution of chickpea-based milk analogues with TG addition (**A**), without TG addition (**B**), and in commercial soymilk (**C**) over 28 days of storage time at 4 °C.

**Figure 3 foods-14-00514-f003:**
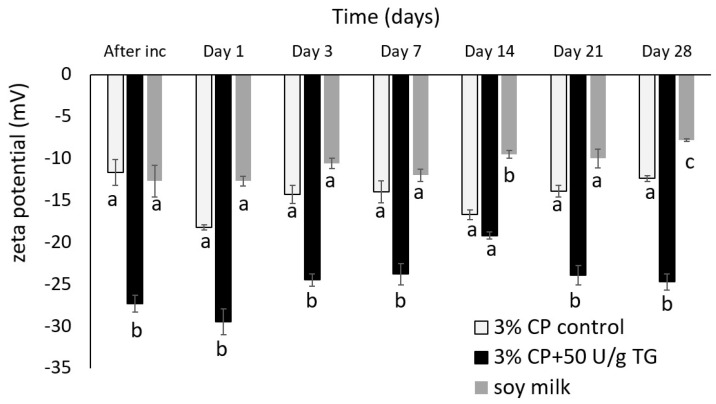
Zeta potential of chickpea-based milk analogues without TG (3% CP control), with TG addition (3% CP + 50 U/g TG), and commercial soymilk sample over the storage time of 28 days at 4 °C. a, b, c different letters denote significant differences (*p* < 0.05) between plant-based milk analogues over one month of storage time at 4 °C.

**Figure 4 foods-14-00514-f004:**
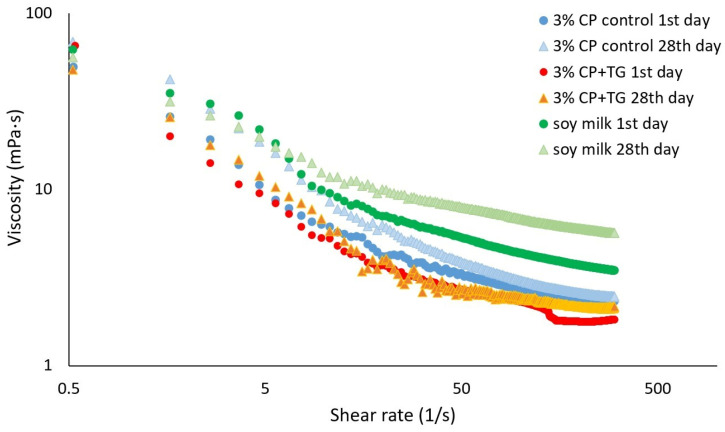
Dynamic viscosity of chickpea-based milk analogues without TG (3% CP control), with TG addition (3% CP + TG), and commercial soymilk sample at 1st and 28th day of storage at 4 °C, as determined at shear rate range of 0.5 to 300 1/s.

**Figure 5 foods-14-00514-f005:**
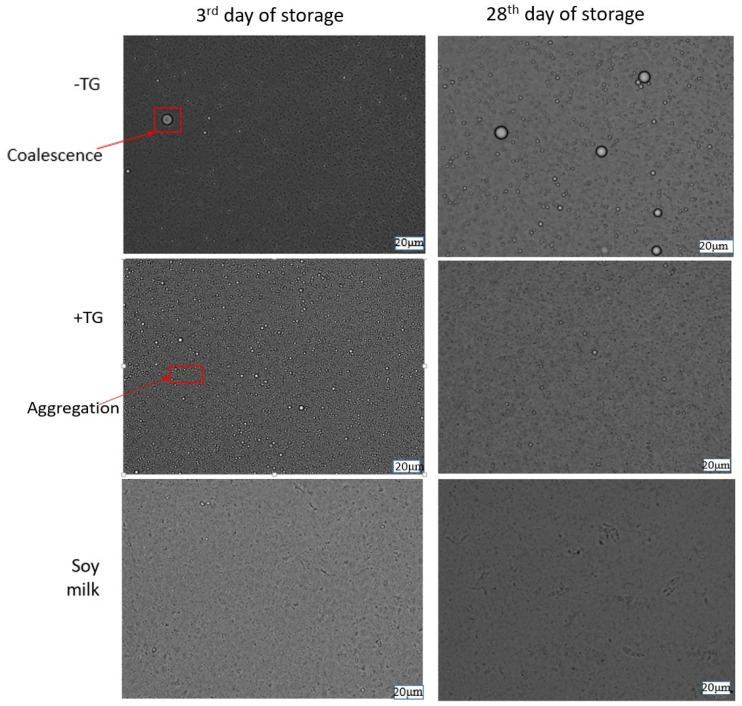
Observation in light microscope (magnitude ×40) of chickpea-based milk analogues without TG addition (-TG), with TG addition (+TG), and commercial soymilk sample at 3rd and 28th day of storage at 4 °C. The red arrows indicate areas of coalescence and aggregation, highlighted within the red squares.

**Table 1 foods-14-00514-t001:** Characteristic droplet size parameters for the different plant-based milk analogues over one month of storage time at 4 °C.

Sample Name	Storage Period	Dx (50)	Dx (10)	Dx (90)	D[4,3] (µm)	D[3,2] (µm)
3% CP Cont.	* After inc.	0.38 ± 0.01 a	0.04 ± 0.01 a	1.95 ± 0.22 a	0.82 ± 0.10 a	0.12 ± 0.01 a
	1st day	0.36 ± 0.01 a	0.04 ± 0.01 a	2.05 ± 0.23 a	0.82 ± 0.05 a	0.12 ± 0.02 a
	3rd day	0.37 ± 0.01 a	0.04 ± 0.01 a	2.66 ± 0.69 a	0.96 ± 0.15 b	0.12 ± 0.01 a
	7th day	0.35 ± 0.01 a	0.04 ± 0.01 a	2.03 ± 0.32 a	0.78 ± 0.07 a	0.11 ± 0.01 ac
	14th day	0.33 ± 0.01 a	0.04 ± 0.01 a	1.63 ± 0.02 a	0.67 ± 0.01 a	0.11 ± 0.01 ac
	21st day	0.60 ± 0.09 b	0.05 ± 0.01 a	142.60 ± 19.27 b	33.42 ± 9.08 c	0.15 ± 0.01 a
	28th day	0.51 ± 0.04 c	0.05 ± 0.01 a	142.00 ± 16.52 b	26.83 ± 4.85 c	0.14 ± 0.01 a
3% CP + TG	* After inc.	0.39 ± 0.01 a	0.03 ± 0.01 a	1.96 ± 0.03 a	0.85 ± 0.01 a	0.12 ± 0.01 a
	1st day	0.39 ± 0.01 a	0.04 ± 0.01 a	2.03 ± 0.01 a	0.87 ± 0.01 a	0.11 ± 0.01 ac
	3rd day	0.42 ± 0.01 a	0.04 ± 0.01 a	1.89 ± 0.01 a	0.86 ± 0.02 a	0.13 ± 0.01 a
	7th day	0.39 ± 0.003 a	0.04 ± 0.01 a	1.74 ± 0.04 a	0.79 ± 0.02 a	0.12 ± 0.02 a
	14th day	0.40 ± 0.003 a	0.04 ± 0.01 a	1.89 ± 0.02 a	0.86 ± 0.01 a	0.13 ± 0.01 a
	21st day	0.45 ± 0.01 a	0.05 ± 0.01 a	4.92 ± 1.93 c	10.53 ± 2.07 d	0.14 ± 0.01 a
	28th day	0.54 ± 0.03 c	0.06 ± 0.01 a	120.87 ± 20.72 b	22.43 ± 0.01 e	0.17 ± 0.01 b
Soymilk	** After op.	0.17 ± 0.01 d	0.04 ± 0.01 a	0.77 ± 0.01 d	0.72 ± 0.04 a	0.09 ± 0.01 c
	1st day	0.17 ± 0.01 d	0.04 ± 0.01 a	0.64 ± 0.01 d	0.43 ± 0.01 f	0.09 ± 0.01 c
	3rd day	0.17 ± 0.01 d	0.04 ± 0.01 a	0.63 ± 0.01 d	0.42 ± 0.01 f	0.09 ± 0.01 c
	7th day	0.16 ± 0.01 d	0.03 ± 0.01 a	0.56 ± 0.01 d	0.31 ± 0.01 f	0.09 ± 0.01 c
	14th day	40.89 ± 0.01 e	0.06 ± 0.01 a	301.50 ± 0.01 e	105.43 ± 0.01 g	0.19 ± 0.01 b
	21st day	4.54 ± 0.42 f	0.05 ± 0.01 a	176.80 ± 14.96 f	53.70 ± 4.061 h	0.17 ± 0.01 b
	28th day	0.58 ± 0.09 b	0.04 ± 0.01 a	202.30 ± 14.57 f	56.70 ± 5.01 h	0.12 ± 0.01 a

* after incubation time, ** after opening; a, b, c, d, e, f, g, h values associated with different letters denote significant differences (*p* < 0.05) between plant-based milk analogues over one month storage time at 4 °C.

## Data Availability

The original contributions presented in this study are included in the article. Further inquiries can be directed to the corresponding author.

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
