# Peer review of "Chickpea-Based Milk Analogue Stabilized by Transglutaminase"

_foods, 2025, doi:10.3390/foods14030514_

Round 1
Reviewer 1 Report
Comments and Suggestions for Authors
The submission reports on crosslinking chickpea proteins using a bacterial transglutaminase to stabilise an emulsion resembling milk. There are several issues with this submission - first of all, nothing is known about chickpea protein isolate (CPPI), and while the authors mentioned that the protein content was determined, it wasn't even reported in the text. Furthermore, the CPPI is actually a mixture of different proteins, and it would be imperative to have a better understanding of the protein preparation. If the aim was to improve the stability of an emulsion resembling milk, why was milk not used as a control? Instead, the authors opted to use soy milk, for which they provided no information. Also, if they wanted to create a system resembling milk, why not create such an environment then, i.e. at least mineral content should be similar and so on? I've included some additional comments in the document I've attached. For this work to be considered for publication, the authors must provide much more information than this submission contains. Yet the absence of proper control is challenging.

Author Response
Dear Reviewer,  
We thank you for the constructive criticism and for giving us the opportunity to improve the manuscript. We have made efforts to answer the reviewer’s comments and clarify the manuscript.
To enhance clarity and better reflect the study's objective, the title has been revised to “Chickpea-based milk analogue stabilized by transglutaminase”.
The figure legends have been rewritten for improved clarity, and new references have been added. Figure 1 has been corrected and replaced, while statistics were added to Figure 3. New Table 1 was added. The manuscript has been thoroughly revised and rewritten to eliminate repetition as much as possible. All modifications made to the manuscript are highlighted in yellow.
We hereby provide a point-by-point response to the reviewer’s comments (the line number refers to the track-change version of the manuscript):
Comment 1: The submission reports on crosslinking chickpea proteins using a bacterial transglutaminase to stabilise an emulsion resembling milk. There are several issues with this submission - first of all, nothing is known about chickpea protein isolate (CPPI), and while the authors mentioned that the protein content was determined, it wasn't even reported in the text. Furthermore, the CPPI is actually a mixture of different proteins, and it would be imperative to have a better understanding of the protein preparation.
Response 1:
We thank the reviewer for this comment. The chickpea protein content was determined, and it was added to manuscript (Line 119). The expected protein yield from raw chickpeas exceeds 80%, which is considered a good yield achieved through isoelectric precipitation.
Chickpea protein consists of mostly of globulin fractions (legumin and vicilin) albumin (enzymatic proteins, protease inhibitors, amylase inhibitors and lectins), prolamin (alcohol soluble; 2.8%), glutelin (acid/alkali soluble; 18.1%). Among chickpea protein fractions, vicilin with a molecular weight of 50 kDa, chickpea α- and β-legumin, lipoxygenase (∼92 kDa) and convicilin (70 kDa) were crosslinked rapidly by TG as shown in our previous study. Please see the Lines 262-266.
Comment 2: If the aim was to improve the stability of an emulsion resembling milk, why was milk not used as a control? Instead, the authors opted to use soy milk, for which they provided no information. Also, if they wanted to create a system resembling milk, why not create such an environment then, i.e. at least mineral content should be similar and so on?
Response 2: We thank the reviewer for highlighting this point. The objective of our study was to develop a chickpea protein-based milk analogue and compare it to another well-established plant-based milk analogue, such as soy milk, rather than directly to dairy milk. To address this and provide greater clarity, we have also revised the manuscript title to “Chickpea-Based Milk Analogue Stabilized by Transglutaminase.”
Information for commercial soymilk was added in the Material section, Lines 97-99: Soy drink lite, Tnuva alternative, 3.3% protein, 1.6% fat, sugar, dietary supplement: calcium carbonate (of which calcium 0.1%) , acidity regulators: [( E-450(v), E-452(i), E-500(i)], flavorings, salt, stabilizer: Gellan Gum, vitamin B12, B2 and D (B2 - 0.24 mg, B12 - 0.4 mcg, D - 0.75 mcg per 100 g)
Comments 3: I've included some additional comments in the document I've attached. For this work to be considered for publication, the authors must provide much more information than this submission contains. Yet the absence of proper control is challenging.
Response 3: The answers to comments are added in the attached document as well. The control in the manuscript is TG non-treated chickpea-based milk analogue, as well as commercial soymilk sample. The manuscript is revised and improved. Please see the new version.
Below is an attached pdf with answers to reviewer comments.

Reviewer 2 Report
Comments and Suggestions for Authors
The authors studied the effect of TG on the o/w emulsion while creating a plant based beverage using chick pea protein and compared with a commercially available soy beverage. Overall a good experimental design and well written manuscript. Bellow improvement suggestions could be considered to improve the readability and the quality of the manuscript.
- The abstract could be added with brief results from the experiment in addition to key findings statements.
- There are no significant differences between the PSD plots of TG-cross linked and non-cross linked chickpea milk. Where as the physical observations showed creaming after 7th day at RT. These inconsistencies could be addressed well.
- Also, the PDS plots do not indicate the storage temperature. Please present the plot of RT if conducted for the benefit of the readers.
- The other reason for the differences in the PDS results between chickpea and soy milks could be the way they are processed. Ideally the soy milk should have been treated using the same conditions utilised for making the chick pea milk to avoid any misleading conclusions.
- Another contradicting finding could be explained/discussed better - the significantly higher zeta potential values in the case of TG-crosslinked treatment is correlating with the PDS comparative results.
Author Response
Dear Reviewer,  
We thank you for the constructive criticism and for giving us the opportunity to improve the manuscript. We have made efforts to answer the reviewer’s comments and clarify the manuscript.
To enhance clarity and better reflect the study's objective, the title has been revised to “Chickpea-based milk analogue stabilized by transglutaminase”.
The figure legends have been rewritten for improved clarity, and new references have been added. Figure 1 has been corrected and replaced, while statistics were added to Figure 3. New Table 1 was added. The manuscript has been thoroughly revised and rewritten to eliminate repetition as much as possible. The sections addressing repetitive comments can be found at the end of this text (in red). All modifications made to the manuscript are highlighted in yellow.
We hereby provide a point-by-point response to the reviewer’s comments (the line number refers to the track-change version of the manuscript):
Comments 1: The authors studied the effect of TG on the o/w emulsion while creating a plant based beverage using chick pea protein and compared with a commercially available soy beverage. Overall a good experimental design and well written manuscript. Bellow improvement suggestions could be considered to improve the readability and the quality of the manuscript.
The abstract could be added with brief results from the experiment in addition to key findings statements.
Response 1: This has been corrected. Please see the revised version.
Comments 2: There are no significant differences between the PSD plots of TG-cross linked and non-cross linked chickpea milk. Where as the physical observations showed creaming after 7th day at RT. These inconsistencies could be addressed well.
Response 2: The PSD was not analyzed in samples stored at RT.
Comments 3: Also, the PDS plots do not indicate the storage temperature. Please present the plot of RT if conducted for the benefit of the readers.
Response 3: The PSD was determined at 4°C as indicated in Figure legend. The samples were stored at room temperature just for visual stability observation.
Comments 4: The other reason for the differences in the PDS results between chickpea and soy milks could be the way they are processed. Ideally the soy milk should have been treated using the same conditions utilised for making the chick pea milk to avoid any misleading conclusions.
Response 4: We agree with the reviewer. To clarify, we have revised this section, and added a Table (Table 1) to enrich the results and discussion. please see L 200-214, and L300-309.
Comments 5: Another contradicting finding could be explained/discussed better - the significantly higher zeta potential values in the case of TG-crosslinked treatment is correlating with the PDS comparative results.
Response 5: Zeta potential and particle size are closely connected, as both play a critical role in the stability of colloidal systems like emulsions. The significantly higher zeta potential values observed in the TG-crosslinked treatment indicate stronger electrostatic repulsion between particles. This enhanced repulsion helps to prevent particle aggregation, leading to a more uniform and stable particle size distribution (PSD). The comparative PSD results align with this, as the TG-crosslinked treatment demonstrates reduced coalescence and aggregation, contributing to improved emulsion stability.
Please read the revised discussion.
Reviewer 3 Report
Comments and Suggestions for Authors
Chickpea-based milk stabilized by transglutaminse paper is interesting, bit some parts some parts of the article require addition and expansion.
Figure 2 It is difficult the there was no difference in particle distribution: samples TG-crosslinked and non-crosslinked chickpea-based milk-like emulsions, over storage. The samples stored had a wider size distribution. Also I suggest to add the table thee average vales and standard deviation with the size of particles (the average value, D32, etc.).
-Please add the statistic (homogenic groups) in Figure 3
-line 191-202 for better comparison of the curves I suggest to use the models describe the changes during the shearing. Based on the vales of K and n give more information. Because the behaviour is non-Newtonian, you should use apparent viscosity in Figure 4.
-In the discussion, the authors should determine the relationships between the parameters studied and, if necessary, explain them.
-Conclusions are statements only, not conclusions. It needs improvement.
Author Response
Dear Reviewer, 
We thank you for the constructive criticism and for giving us the opportunity to improve the manuscript. We have made efforts to answer the reviewer’s comments and clarify the manuscript.
To enhance clarity and better reflect the study's objective, the title has been revised to “Chickpea-based milk analogue stabilized by transglutaminase”.
The figure legends have been rewritten for improved clarity, and new references have been added. Figure 1 has been corrected and replaced, while statistics were added to Figure 3. New Table 1 was added. The manuscript has been thoroughly revised and rewritten to eliminate repetition as much as possible. The sections addressing repetitive comments can be found at the end of this text (in red). All modifications made to the manuscript are highlighted in yellow.
We hereby provide a point-by-point response to the reviewer’s comments (the line number refers to the track-change version of the manuscript):
Comment 1: Chickpea-based milk stabilized by transglutaminse paper is interesting, bit some parts some parts of the article require addition and expansion.
Figure 2 It is difficult the there was no difference in particle distribution: samples TG-crosslinked and non-crosslinked chickpea-based milk-like emulsions, over storage. The samples stored had a wider size distribution. Also I suggest to add the table thee average vales and standard deviation with the size of particles (the average value, D32, etc.).
Response 1: The table has been added as suggested and the results and discussion section have been revised and improved.
Comments 2: Please add the statistic (homogenic groups) in Figure 3
Response 2: The figure has been corrected and replaced.
Comments 3: -line 191-202 for better comparison of the curves I suggest to use the models describe the changes during the shearing. Based on the vales of K and n give more information. Because the behaviour is non-Newtonian, you should use apparent viscosity in Figure 4.
Response 3: The apparent viscosity term was used as suggested. Although the power law could offer deeper insights into the flow behavior of plant-based milk analogues, we believe that the viscosity chart provides sufficient information for this study.
Comment 4: In the discussion, the authors should determine the relationships between the parameters studied and, if necessary, explain them.
Response 4: The discussion has been revised. Please see the revised version of the manuscript.
Comment 5: Conclusions are statements only, not conclusions. It needs improvement.
Response 5: The conclusions have been rewritten. Please see the revised version.
Round 2
Reviewer 1 Report
Comments and Suggestions for Authors
The authors have done a fine job improving their manuscript to the point where it now makes more sense. Probably, they should have included some additional methods, but the discussion and conclusions are now based on the results presented. Some improvements in English expressions are warranted, but scientifically, the submission is sound.
Author Response
We sincerely thank the reviewer for their valuable comments, which have greatly helped us improve the quality of the paper.
Reviewer 3 Report
Comments and Suggestions for Authors
Thee article was improved. It can be accepted.
Author Response

(The authors gave the same response as above.)
